# Skin Surface Dose for Whole Breast Radiotherapy Using Personalized Breast Holder: Comparison with Various Radiotherapy Techniques and Clinical Experiences

**DOI:** 10.3390/cancers14133205

**Published:** 2022-06-30

**Authors:** Chiu-Ping Chen, Chi-Yeh Lin, Chia-Chun Kuo, Tung-Ho Chen, Shao-Chen Lin, Kuo-Hsiung Tseng, Hao-Wen Cheng, Hsing-Lung Chao, Sang-Hue Yen, Ruo-Yu Lin, Chen-Ju Feng, Long-Sheng Lu, Jeng-Fong Chiou, Shih-Ming Hsu

**Affiliations:** 1Department of Radiation Oncology, Wan Fang Hospital, Taipei Medical University, Taipei 110, Taiwan; chiuping@w.tmu.edu.tw (C.-P.C.); 97023@w.tmu.edu.tw (C.-Y.L.); b8601093@tmu.edu.tw (C.-C.K.); 109088@w.tmu.edu.tw (H.-L.C.); sanghueyen@gmail.com (S.-H.Y.); 2Department of Biomedical Imaging and Radiological Sciences, National Yang Ming Chiao Tung University, Taipei 112, Taiwan; dorislin@gm.ym.edu.tw (R.-Y.L.); iris450399@hotmail.com (C.-J.F.); 3Department of Radiation Oncology, Taipei Medical University Hospital, Taipei Medical University, Taipei 110, Taiwan; tungho_chen@yahoo.com.tw (T.-H.C.); 123007@h.tmu.edu.tw (L.-S.L.); 4School of Health Care Administration, College of Management, Taipei Medical University, Taipei 110, Taiwan; 5Ph.D. Program for Cancer Molecular Biology and Drug Discovery, College of Medical Science and Technology, Taipei Medical University and Academia Sinica, Taipei 110, Taiwan; 6Graduate Institute of Biomedical Materials and Tissue Engineering, College of Biomedical Engineering, Taipei Medical University, Taipei 110, Taiwan; sa59804108lly@gmail.com; 7Department of Electrical Engineering, National Taipei University of Technology, Taipei 10608, Taiwan; f10473@mail.ntut.edu.tw; 8Department of Radiation Oncology, Shuang Ho Hospital, Taipei Medical University, Taipei 11031, Taiwan; haowen1203@gmail.com; 9School of Biomedical Engineering, College of Biomedical Engineering, Taipei Medical University, Taipei 11031, Taiwan; 10Department of Radiation Oncology, Tri-Service General Hospital, National Defense Medical Center, Taipei 11490, Taiwan; 11TMU Research Center of Cancer Translational Medicine, Taipei Medical University, Taipei 110, Taiwan; 12International Ph.D. Program in Biomedical Engineering, College of Biomedical Engineering, Taipei Medical University, Taipei 110, Taiwan; 13International Ph.D. Program for Cell Therapy and Regenerative Medicine, College of Medicine, Taipei Medical University, Taipei 110, Taiwan; 14Department of Radiology, School of Medicine, College of Medicine, Taipei Medical University, Taipei 110, Taiwan

**Keywords:** breast cancer, radiotherapy, intensity-modulated radiotherapy, personalized breast holder (PERSBRA), surface dose, volumetric modulated arc therapy

## Abstract

**Simple Summary:**

Breast immobilization with personalized breast holder (PERSBRA) is a promising approach to reduce the toxicity in the lungs and heart during whole breast radiotherapy. In this study, we designed PERSBRA with three different mesh sizes (large, fine and solid) and applied them on a Rando phantom. Hybrid, intensity modulated radiation therapy (IMRT), and volumetric modulated arc therapy (VMAT) techniques were used to deliver a prescribed dose. The dose measurement with EBT3 film and TLD were taken on Rando phantom with no PERSBRA, large mesh, fine mesh, and solid PERSBRA for tumor doses and surface doses. This innovative PERSBRA provides another radiotherapy option for patients with left breast cancer.

**Abstract:**

Purpose: Breast immobilization with personalized breast holder (PERSBRA) is a promising approach for normal organ protection during whole breast radiotherapy. The aim of this study is to evaluate the skin surface dose for breast radiotherapy with PERSBRA using different radiotherapy techniques. Materials and methods: We designed PERSBRA with three different mesh sizes (large, fine and solid) and applied them on an anthropomorphic(Rando) phantom. Treatment planning was generated using hybrid, intensity-modulated radiotherapy (IMRT) and volumetric-modulated arc therapy (VMAT) techniques to deliver a prescribed dose of 5000 cGy in 25 fractions accordingly. Dose measurement with EBT3 film and TLD were taken on Rando phantom without PERSBRA, large mesh, fine mesh and solid PERSBRA for (a) tumor doses, (b) surface doses for medial field and lateral field irradiation undergoing hybrid, IMRT, VMAT techniques. Results: The tumor dose deviation was less than five percent between the measured doses of the EBT3 film and the TLD among the different techniques. The application of a PERSBRA was associated with a higher dose of the skin surface. A large mesh size of PERSBRA was associated with a lower surface dose. The findings were consistent among hybrid, IMRT, or VMAT techniques. Conclusions: Breast immobilization with PERSBRA can reduce heart toxicity but leads to a build-up of skin surface doses, which can be improved with a larger mesh design for common radiotherapy techniques.

## 1. Introduction

In 2020, breast cancer became the most common cancer in women worldwide, accounting for 11.7% of female cancer cases [1]. The five-year survival rate for breast cancer increased from 75% (1975–1977) to 90% (2002–2008) [2]. The five-year survival rate for early-stage breast cancer is up to 95% due to early diagnosis of breast cancer with the prevalence of breast cancer screening. As there are no differences in disease-free survival or overall survival between mastectomy and breast preservation therapy in patients with early-stage breast cancer [3], 70–80% of patients with early-stage breast cancer choose breast preservation therapy for aesthetic reasons [4]. Adjuvant whole-breast radiation therapy after breast preservation surgery can reduce the rate of local recurrence [5]. However, radiation-induced complications are one of the main concerns, such as acute skin discoloration due to toxicity. Additionally, radiation pneumonitis can occur two to three months after radiation therapy in one percent of patients, and lymphedema can occur a few weeks to a few years after radiation therapy or even later for the occurrence of cardiac disease [6].

In patients with left breast cancer who receive whole breast radiotherapy, the additional radiation dose received by organs at risk (lungs, heart) must also be considered in addition to tumor coverage. Previous studies have shown that the rates of major coronary events and cardiotoxicity increased with mean heart dose by 7.4% per Gy [7]. Furthermore, each increase in the mean dose to the heart by 1 Gy increases the risk of cardiotoxicity by four percent [8]. There are two common heart-sparing radiotherapy techniques, and the first is deep inspiration breath hold (DIBH) [9,10,11,12,13,14,15,16]. DIBH can increase the distance from the heart to the treatment field due to lung expansion, resulting in a significant reduction in the mean and maximum doses to the heart and left ventricle [17]. Breast irradiation in the prone position is another heart-saving radiotherapy technique [18]. Previous studies have shown that the prone position can significantly reduce lung and heart doses for patients with large breast volumes (>896 mL) [19,20], because the prone position can move the treatment field away from the heart. Krengli et al. reported no statistically significant differences in mean heart dose between the prone and supine positions, but heart V5 (VX, the percentage volume that receives at least X Gy) and V10 in the prone position were lower [21].

Although DIBH is an effective technique to preserve the heart and lung in left breast cancer radiotherapy [22], patients must hold their breath for at least 30 s to be eligible for DIBH. In terms of the prone technique, patients often complain of neck and spinal pain due to maintaining the prone position [20]. Therefore, the main challenge in prone position breast irradiation is to maintain the correct position with the treated breast hanging away from the treatment field in a comfortable and reproducible way. In this study, a new personalized breast holder (PERSBRA) was designed that allows patients to be treated more comfortably and reproducibly in the supine position. PERSBRA, as a breast support and immobilization device, can increase the distance from the heart to the treatment field, and it allows patients to be treated in a more comfortable and reproducible supine position.

Radiotherapy techniques for breast cancer have progressed from the traditional three-dimensional (3D) conformal radiotherapy (3D-CRT) technology to two opposed tangent photon fields [23,24,25,26,27,28] to intensity-modulated radiotherapy (IMRT) [29] and to volumetric modulated arc therapy (VMAT) [30]. Several studies have compared doses to organs at risk (OARs) among 3D-CRT, IMRT, and VMAT techniques for breast cancer [30,31,32,33,34,35,36]. For left breast cancer, IMRT and VMAT techniques can achieve better dose homogeneity in the target and avoidance of OAR (lung and heart) [37]. Currently, these three treatment techniques are used in clinical breast cancer radiotherapy.

PERSBRA simulates the patient in the prone position. After increasing the distance between the heart and the breast to be irradiated, the PERSBRA maintains the reproducibility of the patient in the supine position. The new personalized breast holder, PERSBRA, would be thicker than the thermoplastic mask used in clinical routine to immobilize the breast in the supine position to maintain the distance from the heat to the treatment field. The purpose of this study was to evaluate the application of this innovative PERSBRA in three clinical breast cancer radiotherapy techniques (Hybrid, IMRT, and VMAT) for left breast cancer using different designs of PERSBRAs, and to evaluate the feasibility of PERSBRA. Based on the results of this study, the feasibility of PERSBRA for breast cancer radiotherapy can be evaluated accordingly.

## 2. Materials and Methods

### 2.1. PERSBRA Design

#### 2.1.1. Rando Phantom

The design method used in this experiment was the same as that used to obtain the breast contours of the patients in our clinic. The contours of both breasts were scanned from the female Rando phantom, and a 3D printer was used to print PERSBRA. The three-dimensional printing of PERSBRA required high-elasticity and biocompatibility plastic filaments. Therefore, we used thermoplastic elastomer (TPE) as the PERSBRA material. The interior of the PERSBRAs consisted of a hollow honeycomb structure, distinguished by TPE filaments of different diameters; filaments with smaller diameters resulted in larger pores. Three PERSBRAs with different meshes were designed (Figure 1): a 0.35 cm diameter large mesh PERSBRA, a 0.45 cm diameter fine mesh PERSBRA, and a solid PERSBRA. The primary purpose of the PERSBRA was to support breast positioning. Part of the upper portion of the PERSBRA was removed to form a semi-mask-type breast fixation mold, which not only ensured the reproducibility of the patient’s treatment position each time but also prevented a bolus effect [38].

#### 2.1.2. Patient

Figure 2 shows how breast cancer patients were fitted with PERSBRA. The patient is told to stand with their feet shoulder width apart, raise both hands and place them in front of the treatment couch. In this position, the back is naturally straightened, the two arms are extended, and the breasts are allowed to hang naturally, placing the upper half of the body in the prone position. An infrared scanner was used to obtain the contours of the entire left and right breasts. The patient’s breast contour data was then transferred to a dedicated image processing system (The breast contour image was exported as an STL file by Skanect software version 1.9 (Occipital, San Francisco, CA, USA). Then the customized PERSBRA was designed by the Meshmixer program), and a personalized device for the patient was fabricated using 3D printing technology. Depending on the size of the breast of each patient, it took about 18 to 40 h to make the 3D print of PERSBRA. Before CT simulation, each patient must spend about 5 min to acquire the 3D breast shape by 3D scanner. Finally, it took only one minute to wear PERSBRA before treatment.

### 2.2. Simulation and Planning of Treatment

#### 2.2.1. Rando Phantom

The Rando phantom underwent computed tomography (CT) scans with Brilliance CT Big BoreTM (Philips, Amsterdam, The Netherlands) without PERSBRA (a), with large-mesh PERSBRA (b), with fine-mesh PERSBRA (c) and with solid PERSBRA (d). The clinical target volume (CTV) and the OARs (heart and lung) were delineated on these CT scans.

Three different treatment plans (Hybrid, IMRT, VMAT) were developed. All plans were generated with 6-MV X-ray photon beams using a Pinnacle treatment planning system(TPS) (Philips, version 9.8C, Fitchburg, WI, USA). The hybrid whole-breast treatment plan was based on a combination of 80% of the 3D-CRT field plus 20% of 2–3 IMRT fields (Figure 3a). Tangential fields were used for 3D-CRT. The medial tangential field was 310° to the phantom and the lateral tangential field 130° to the phantom. The field angle for IMRT was the tangential field elevated to about 10–20°. The dose conformity of the hybrid technique was superior to that of the traditional two-tangential field (3D-CRT) technique. Furthermore, 5–6 IMRT treatment fields were designed according to the shape and size of the breast (Figure 3b) in order to reduce the radiation dose to the surrounding normal tissue. The difference between all IMRT treatment fields and the hybrid was that all of the original tangential fields were changed to IMRT treatment fields. In this study, the surface dose with the latest VMAT treatment technology using PERSBRA was also evaluated and two partial arc rotation angle treatments, from 310° clockwise to 180°, and then 180° counterclockwise to 310° (Figure 3c) were designed accordingly.

The treatment planning for hybrid, IMRT, and VMAT was based on clinical guidelines for total breast radiation therapy [39], applying 200 cGy to the whole breast (CTV) in 25 fractions for a total dose of 5000 cGy. For the evaluation of the treatment plan, 95% of the prescribed dose had to cover at least 95% of the tumor volume (CTV), and the dose to normal tissue had to be as low as possible.

#### 2.2.2. Patient

A total of 25 patients with left breast cancer were enrolled in this clinical trial (IRB: TMU-N201603037). Two sets of CT images were collected from each patient in the clinical trial: one without PERSBRA and another with PERSBRA. The physician delineated the CTV and surrounding OARs (contralateral breast, lung, heart, and left anterior descending artery) from the CT of the two groups. The planning target volume (PTV) was a 5 mm isotropic expansion of the CTV due to the influence of the uncertainty of the setup and the respiratory motion. Based on clinical guidelines for whole breast radiotherapy [39], the prescription dose for all patients was set at 5000 cGy in 25 fractions. For evaluation of treatment planning, 95% of the prescribed dose had to cover at least 95% of the planning target volume (PTV) as specified by the Radiation Therapy Oncology Group (RTOG) [40], and the dose to the OARs had to be restricted to V20 ≤ 25% for the ipsilateral lung and V25 ≤ 10% of the heart, keeping the dose to LAD as low as possible [37]. The 25 patients underwent hybrid radiotherapy technique (combined 80% 3DCRT and 20% IMRT) with solid PERSPRA according to the RTOG treatment guideline.

### 2.3. Surface Dose Measurement

#### 2.3.1. Rando Phantom

In this experiment, GafChromic EBT3 films and ultrathin thermoluminescent dosimeters (TLD, GR-200F, surface area 0.5 × 0.5 cm^2^, nominal thickness 5 mg cm^−2^) were used to measure breast surface doses with each of the three breast cancer treatment techniques (hybrid, IMRT, VMAT) without PERSBRA (a), with the large-mesh PERSBRA (b), with the fine-mesh PERSBRA (c), and with solid PERSBRA (d). The calibration curves of EBT3 films and TLDs were established by a Farmer-type ionization chamber (PTW TW30013) in a 6 MV linear accelerator (Elekta Synergy) at our clinical reference conditions of 1 cGy/MU (SAD = 100 cm, depth = 5 cm, field size = 10 × 10 cm^2^). EBT3 films and TLDs were placed at the depth of our reference conditions to irradiate different doses (50 cGy, 100 cGy, 150 cGy, 200 cGy, 250 cGy, 300 cGy and 350 cGy). The EBT3 films were scanned after 24 h of irradiation and digitized with Epson 11000XL (Epson America Inc., Long Beach, CA, USA). The GR-200F TLD Readout (Rexon components, Inc., beachwood, OH, USA) was performed with a UL-320 TLD reader after 24 h of irradiation.

A reference point (Ref) was selected in the breast tumor CTV to be irradiated to confirm the accuracy of the EBT3 films and the TLD measurement. Furthermore, two positions were selected for the measurement of breast surface dose, with point 1 (P1) in the medial field and point 2 (P2) in the lateral field (Figure 4). A 2 × 2 cm^2^ EBT3 film and TLD were placed in the CTV of the Rando phantom breast, and the EBT3 film and TLD were placed in both the medial and lateral fields on the surface of the breast. Two TLDs were placed at each surface measurement site to measure the reference point and surface doses for each of the three breast cancer treatment techniques (hybrid, IMRT, VMAT) without PERSBRA (a), with the large-mesh PERSBRA (b), with the fine-mesh PERSBRA (c) and with the solid PERSBRA (d). For each technique, measurements were performed in triplicate, and the average was taken to ensure measurement stability and consistency.

#### 2.3.2. Patient

In the human clinical trial, to ensure the effectiveness of breast support, solid PERSBRA was used for each patient. Based on clinical guidelines for whole breast radiotherapy, the prescribed dose was 5000 cGy in 25 fractions. During treatment, the EBT3 film (2 × 2 cm^2^) was placed directly by tape on the skin surface of the patient at P1-anterior, P2-lateral, P3-posteior, and P4-medial direction, which was covered with solid PERSBRA. The surface dose was measured three times in four different positions on the skin surface (Figure 5).

### 2.4. Statistics

The paired Wilcoxon signed rank exact test (Mann–Whitney test) was used for this study. The analysis software was conducted using R version 4.0.3. The level of statistical significance was considered at a *p*-value of < 0.05 for all tests.

## 3. Results

### 3.1. Rando Phantom

Treatment with the three different techniques (hybrid, IMRT, and VMAT) without PERSBRA, with the large-mesh PERSBRA, with the fine-mesh PERSBRA, and with the solid PERSBRA achieved 95% coverage of tumor volume (CTV) with 95% of the prescribed dose. The dose to OARs with solid PERSBRA was as follows: the mean dose of the hybrid treatment = 216.4 cGy; the mean dose of IMRT = 201.9 cGy; and the mean dose of VMAT = 128.1 cGy. In the mean dose of left lung, the hybrid treatment was 571.7 cGy, the mean dose of IMRT was 597.1 cGy and the mean dose of VMAT was 591.3 cGy. However, in the low-dose lung volume that received 10 Gy (V10), the hybrid V10 was 11.03%, the IMRT V10 was 12.42% and the VMAT V10 was 15.97% (Figure 6). There were five treatment fields for the hybrid and IMRT treatments. The hybrid plans reduced the total monitor units (MU) by up to 17% and the treatment delivery time by approximately a minute deviation compared with the IMRT plans. However, the VMAT plans showed higher total MU (7–36% higher) but shorter treatment delivery times (1–2 min shorter) compared with the hybrid and IMRT plans.

Compared with baseline measurement at the reference points at the tumor, the maximum deviation of the EBT3 films and TLD using the three different techniques was 2.14% and 3.92%, respectively, without PERSBRA (a); −1.91% and 3.76%, respectively, with large-mesh PERSBRA (b); −1.11% and 4.10%, respectively, with fine-mesh PERSBRA (c); and −1.88% and −4.92%, respectively, with solid PERSBRA (d) (Table 1). Our results show that the dose differences between the calculated and measured doses of the treatment plan were all within ± 5% for three different radiotherapy techniques without and with different PERSBRA [41].

For surface dose measurement, the maximum deviation of the EBT3 films and TLD using the three different hybrid techniques, IMRT, and VMAT were 15.0%, 18.6%, and 33.1%, respectively, without PERSBRA; 8.3%, 3.6%, and 7.8%, respectively, with the large-mesh PERSBRA; 2.6%, 1.5%, and 6.0%, respectively, with the fine-mesh PERSBRA; and 2.3%, 1.9%, and 2.2%, respectively, with the solid PERSBRA (Table 2).

### 3.2. Patient

The V10 and V20 of the ipsilateral lung have been reduced from 20.0 (5.0)% and 15.0 (4.0)% for patients without solid PERSBRA to 15.5 (4.0)% and 10.0 (5.0)% for patients with solid PERSBRA. The mean heart dose with solid PERSBRA was reduced from 473.8 (168.0) cGy to 335 (144.1) cGy, the mean heart dose with PERSBRA was reduced by 29.3%, the mean LAD dose was reduced from 2021.9 (918.5) cGy to 1433.8 (868.8) cGy, and the mean LAD dose using PERSBRA was reduced by 29.1% (Table 3). The mean contralateral breast doses of the 25 patients were 58.8 (24.2) cGy with solid PERSBRA, 64.5 (38.5) cGy without PERSBRA. Stovall M. et al. have reported that the contralateral breast dose <100 cGy did not increase secondary malignancy [42]. The actual measurement of the patient’s surface dose (Figure 7) showed that the surface dose of TPS was underestimated by 7.3–30% compared to the measurement of the EBT3 film.

## 4. Discussion

For the dose at the reference point on the CTV, the EBT3 film and the TLD results show that the dose deviation between the calculated and measured doses in the planning was within ± 5% for three radiotherapy techniques without and with different mesh PERSBRA (Table 1) [41].

For whole breast radiation therapy using high-energy photon beams (6 MV), the effect of electronic disequilibrium in the build-up region causes a large variation in surface dose. Previous studies have investigated surface dose measurements [43,44,45]. The calculated and measured surface doses for the Rando phantom are shown in Table 2. The three different techniques showed that there were no significant differences (*p* > 0.05) in surface doses (P1 and P2) between the EBT3 film and the TLD measurements for the Rando phantom with different PERSBRAs. The PERSBRAs used in this study were all 5 mm thick; therefore, surface dose measurements were performed in the build-up region, but at a depth with fewer dose variations [46]. Our results also show that the surface dose increased as the mesh size of PERSBRA decreased in the three radiotherapy techniques.

The percentage of depth dose in the build-up region could change from 25% to 50% from depth = 0 mm to depth = 1 mm [46]. It is very difficult to generate precision in superficial dose distribution in the treatment planning system (TPS). Our results (Table 2) show that TPS seems to underestimate surface doses. There are many articles that use Monte Carlo to simulate the surface dose in breast cancer radiotherapy [47,48,49]. Arbor et al. reported a mean difference of 7% for Monte Carlo and 25% for the TPS compared with measurement data in the build-up range [50].

The results of the present study (Table 3) clearly show that the use of solid PERSBRA in all patients with left breast cancer reduced the dose to OARs (lung and heart) by at least 20%; in particular, the mean heart dose was reduced by approximately 29.3%. Therefore, the probability of major cardiovascular events was reduced by approximately 10.3% [7]. This innovative PERSBRA provides another radiotherapy option for patients with left breast cancer who cannot use the DIBH technique.

To avoid skin toxicity caused by the solid PERSBRA, patients were required to apply NS-21 cream (Plunkett Pharmaceuticals, Ltd., Sydney, Australia), a natural cortisone-free cream, after radiation therapy each day. After the whole course of treatment, it was clear that there was only slight Grade 1 erythema of the skin (Figure 8). This was consistent with the findings of Chou et al. that showed that the use of NS-21 cream in patients with head and neck cancer increased skin moisturization [51]. As skin toxicity can develop at different stages after radiotherapy, regular follow-up is required once a week during the course of radiotherapy. The frequency is changed to once a month for three months after the treatment is complete.

Lee et al. found that the head and neck thermoplastic mask is commonly used because the bolus effect increases the surface dose by 18% [52]. The thickness of our PERSBRA is about 5 mm in order to achieve breast fixation. This is thicker than the commonly used thermoplastic mask, resulting in a significant bolus effect. The surface dose measurement results with the phantom (Table 2), the surface dose with the fine mesh PERSBRA, and the solid PERSBRA are similar. Furthermore, the surface dose of large mesh PERSBRA is 6.7–19.2% lower than that of solid PERSBRA. The surface dose to the breast skin can be effectively reduced by reducing the thickness of the PERSBRA.

On the basis of the results of this study, the surface dose increases as the PERSBRA mesh decreases. However, it increases the distance between the heart and the breast and reduces the risk of major cardiovascular events by 10.3%. For patients undergoing total breast radiation therapy, PERSBRA can reduce the radiation dose received by normal tissues of the thoracic cavity if the DIBH technique cannot be used. Mean doses to OARs (left lung, heart, and LAD) for patients with solid PERSBRA were at least 20% lower than those for patients without PERSBRA(Table 3, Figure 9).

The weakness of the study was the small cohort size and insufficient surface dosimetry data on the actual patient. When using PERSBRA, the patients may worry about skin toxicity. In the future, we need to obtain more surface dosimetry data of actual patients and examine the correlation with skin toxicity. In addition, to evaluate surface dose of PERSBRA using Monte Carlo.

## 5. Conclusions

This innovative PERSBRA provides another radiotherapy option in patients with left breast cancer who cannot use the DIBH technique. Large mesh PERSBRA can be recommended for our future clinical use due to the lower surface dose.

## Figures and Tables

**Figure 1 cancers-14-03205-f001:**
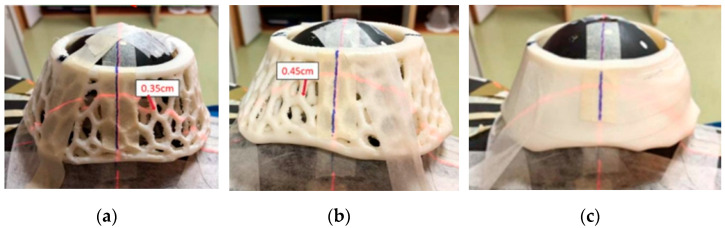
Three different PERSBRA holders. (**a**) Large-mesh PERSBRA; (**b**) fine-mesh PERSBRA; and (**c**) solid PERSBRA.

**Figure 2 cancers-14-03205-f002:**
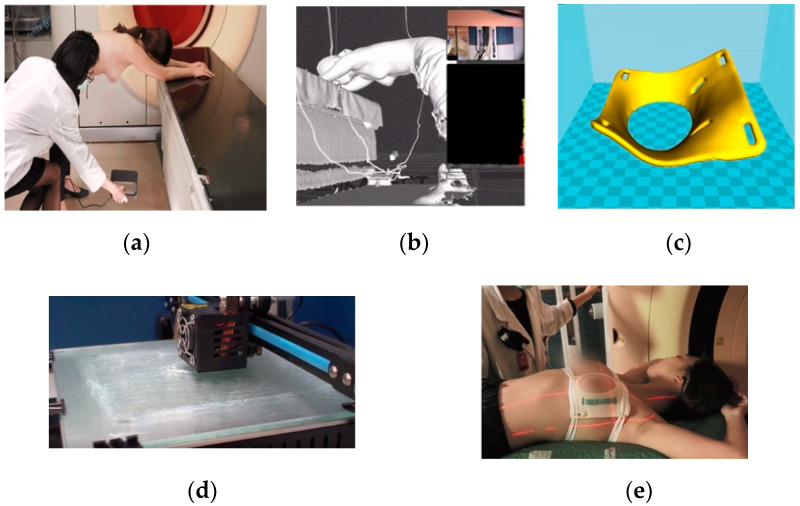
PERSBRA fabrication process: (**a**) breast scan with an infrared scanner; (**b**) 3D breast contour generation; (**c**) 3D PERSBRA model generation; (**d**) 3D printing process for PERSBRA; and (**e**) treatment with PERSBRA.

**Figure 3 cancers-14-03205-f003:**
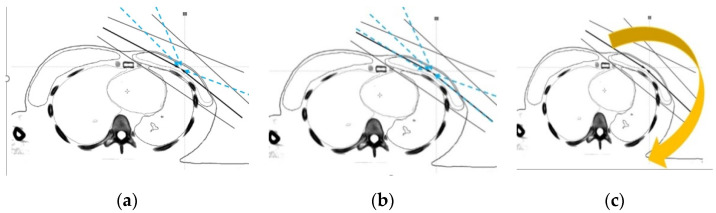
Beam arrangements of different techniques: (**a**) Hybrid consisting of 3D-CRT and IMRT fields; (**b**) five IMRT fields; and (**c**) VMAT with two partial arcs. Abbreviations Hybrid = three-dimensional conformal radiotherapy + intensity modulated radiotherapy; IMRT = intensity modulated radiotherapy; VMAT = volumetric modulated arc therapy.

**Figure 4 cancers-14-03205-f004:**
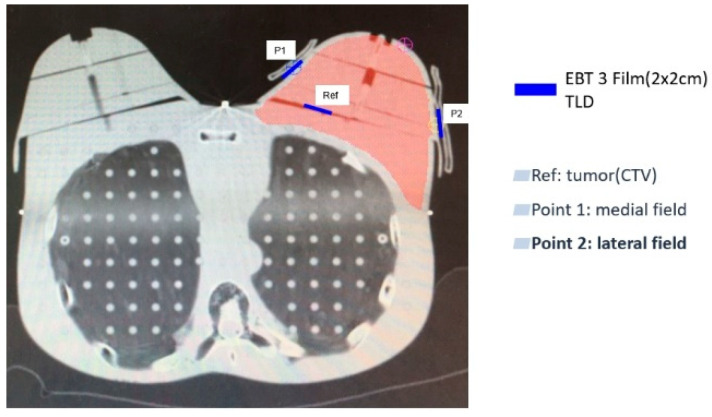
Surface dose measurement points in the Rando phantom. Red represents the clinical target volume (CTV); dark blue represents the measurement points of the EBT3 film and the TLDs. Abbreviations Ref = reference point in the tumor; P1 = surface measurement point one in the medial field; P2 = surface measurement point two in the lateral field.

**Figure 5 cancers-14-03205-f005:**
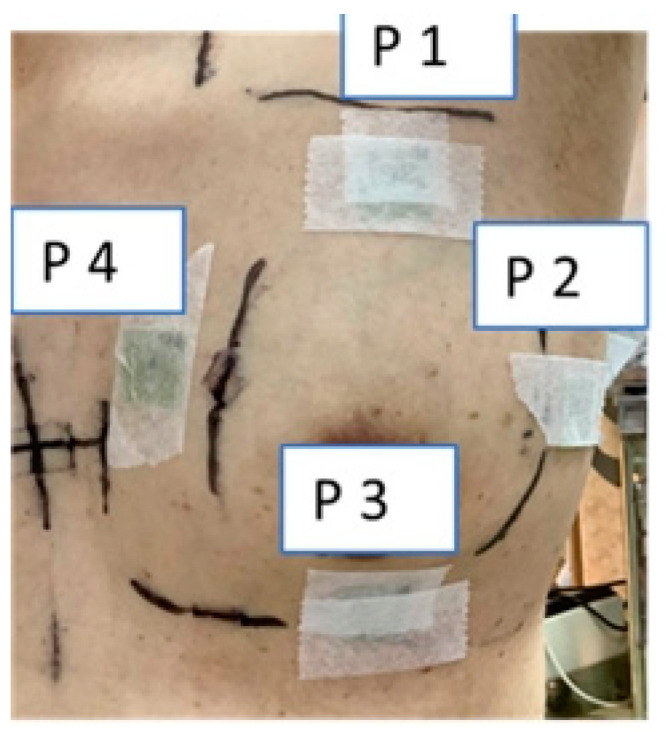
Patient surface dose measurements at four different points (P1–P4).

**Figure 6 cancers-14-03205-f006:**
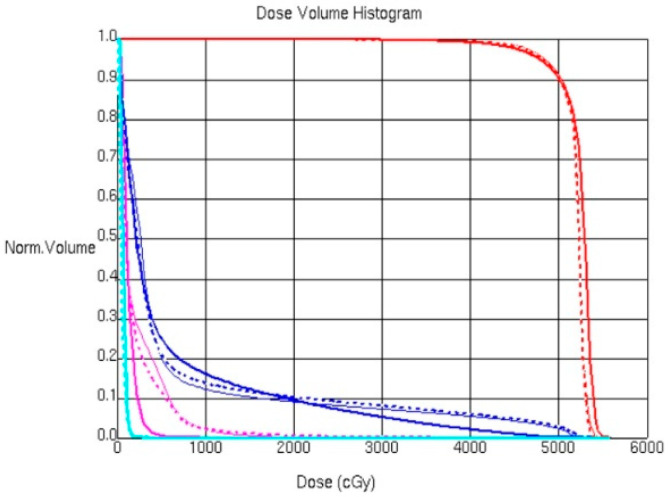
Comparison of CTV dose volume histograms (DVHs) and normal tissues (left lung and heart) for the Rando phantom with solid PERSBRA for three different techniques (hybrid, IMRT, and VMAT). Red represents CTV, blue represents left lung, purple represents heart, thin solid line represents hybrid technique, dashed line represents intensity modulated radiation therapy, and medium solid line represents volumetric modulated arc therapy.

**Figure 7 cancers-14-03205-f007:**
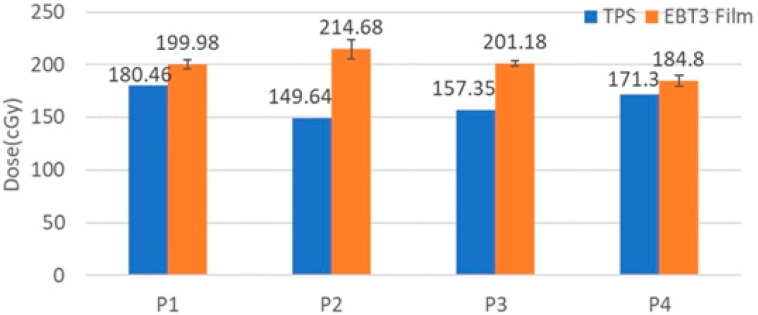
Calculated and measured surface doses for P1–P4 of a representative patient. The error bars represent one standard deviation of the EBT3 film measurements.

**Figure 8 cancers-14-03205-f008:**
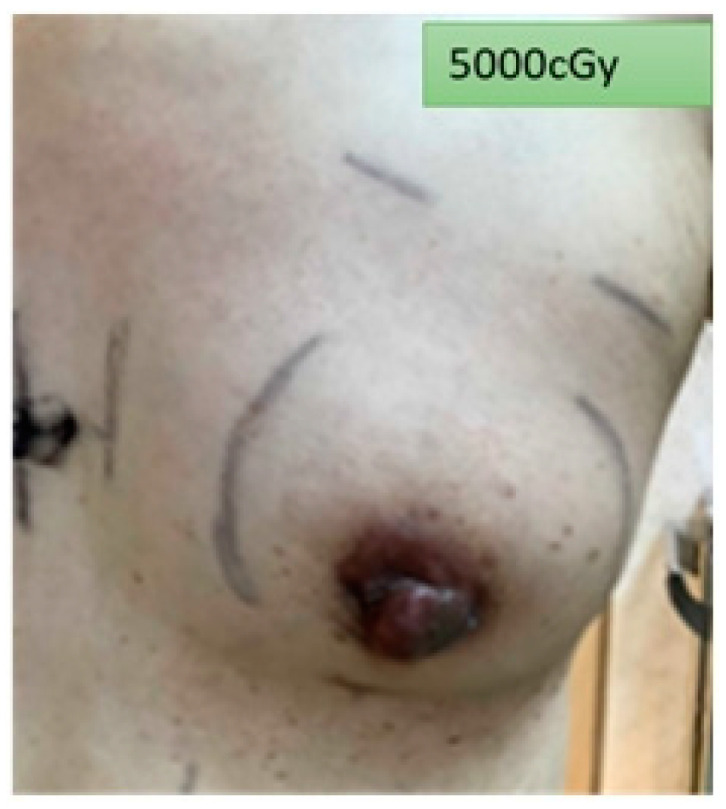
Patient skin reaction after the end of radiotherapy (5000 cGy).

**Figure 9 cancers-14-03205-f009:**
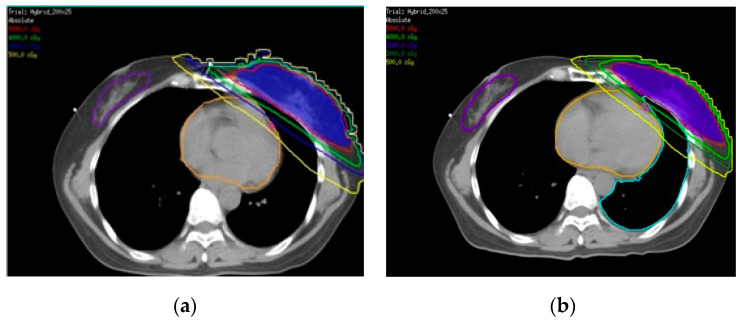
Dose distribution with solid PERSBRA (**a**) and without solid PERSBRA (**b**) for one patient. The red line is the isodose line of 5000 cGy, the blue line is 2000 cGy, and the yellow line is 500 cGy.

**Table 1 cancers-14-03205-t001:** Comparison of calculated and measured doses for planning at the reference point of Rando phantom for three different techniques without and with different PERSBRAs.

Median Dose	Hybrid					IMRT					VMAT				
(cGy)	TPS Baseline	EBT3 Film	*p* Value	TLD	*p* Value	TPS Baseline	EBT3 Film	*p* Value	TLD	*p* Value	TPS Baseline	EBT3 Film	*p* Value	TLD	*p* Value
(IQR)															
Without PERSBRA	210.84	212.14	0.75	215.54	0.5	213.12	218.75	0.5	219.54	0.25	212.52	217.51	0.25	211.23	1
		(3.00)		(10.61)			(4.70)		(3.00)			(2.34)		(3.01)	
Large Mesh PERSBRA	211.33	213.07	0.75	218.46	0.5	211.86	208.55	0.25	217.71	0.75	210.84	206.8	0.25	203.58	0.75
		(3.99)		(13.61)			(0.91)		(15.85)			(0.42)		(17.27)	
Fine Mesh PERSBRA	212.31	216.2	1	214.09	0.75	215.13	224.49	0.25	220.02	0.25	220.55	225.82	0.75	217.23	1
		(6.52)		(5.27)			1.54)		(6.81)			(7.00)		(11.75)	
Solid PERSBRA	210.2	212.59	0.25	224	0.25	210.96	215.52	0.25	205.75	0.25	215.21	209.80	0.5	220.91	0.25
		(1.29)		(3.21)			(2.67)		(4.44)			(5.37)		(6.50)	

Abbreviations TPS: Treatment planning system.

**Table 2 cancers-14-03205-t002:** Comparison of calculated and measured doses at the Rando phantom surface points for three different techniques without and with different PERSBRAs.

	Hybrid								IMRT								VMAT							
Median Dose	P1 (Medial Field)				P2 (Lateral Field)				P1 (Medial Field)				P2 (Lateral Field)				P1 (Medial Field)				P2 (Lateral Field)			
(cGy)	TPS	EBT3 Film	TLD	*p* Value	TPS	EBT3 Film	TLD	*p* Value	TPS	EBT3 Film	TLD	*p* Value	TPS	EBT3 Film	TLD	*p* Value	TPS	EBT3 Film	TLD	*p* Value	TPS	EBT3 Film	TLD	*p* Value
(IQR)																								
Without PERSBRA	25.88	100.34	89.92	0.25	41.44	132.83	112.90	0.25	25.21	106.49	88.29	0.25	40.56	147.20	119.87	0.25	44.93	130.36	100.40	0.25	64.25	149.26	99.86	0.25
		(2.65)	(3.70)			(1.74)	(3.50)			(5.54)	(1.11)			(1.02)	(5.14)			(1.16)	(4.57)			(3.44)	(6.10)	
Large Mesh PERSBRA	92.18	172.08	157.73	0.25	136.55	172.80	171.05	0.25	89.36	163.98	169.94	0.50	131.98	170.58	165.28	0.5	91.51	160.96	148.47	0.25	96.60	159.95	156.16	0.5
		(5.03)	(11.12)			(2.79)	(3.28)			(1.59)	(8.00)			(1.48)	(4.96)			(0.67)	(3.55)			(5.20)	(2.06)	
Fine Mesh PERSBRA	133.56	184.06	179.29	0.75	160.57	194.12	196.76	1	151.84	180.50	183.12	0.25	153.13	202.50	201.66	0.25	157.18	166.08	156.07	0.5	132.74	202.42	194.65	0.25
		(7.67)	(13.24)			(5.28)	(3.98)			(2.38)	(1.55)			(0.90)	(3.90)			(2.77)	(7.13)			(0.75)	(6.47)	
Solid PERSBRA	157.04	184.52	182.15	1	160.24	195.34	199.78	0.25	158.77	181.07	184.51	0.25	159.06	203.06	199.55	1	187.00	178.90	182.90	1	141.18	198.03	199.27	1
		(1.81)	(4.88)			(0.93)	(1.07)			(0.44)	(4.33)			(0.56)	(6.01)			(4.19)	(15.67)			(2.76)	(3.01)	

Abbreviations TPS: Treatment planning system.

**Table 3 cancers-14-03205-t003:** Dosimetric comparison of organs at risk between patients without and with solid PERSBRA.

OARs	without PERSBRA-Hybrid Median (IQR)	with PERSBRA-Hybrid Median (IQR)	*p* Value
Lt Lung V20 (%)	15.0 (4.0)	105.0 (54.0)	<0.0001
Lt Lung V10 (%)	20.0 (5.0)	15.5 (4.0)	<0.0001
Heart Dmean (cGy)	473.8 (1684.0)	335 (144.1)	0.0019
LAD Dmean (cGy)	2021.1 (918.5)	1433.8 (868.8)	0.0022
Rt breast Dmean (cGy)	58.8 (24.2)	64.5 (38.5.2)	0.1116
All data (N = 25) are presented as median (IQR)			

Abbreviations Dmean = mean dose (cGy); LAD = left anterior descending artery; Vx = volume (%) receiving x dose (Gy).

## Data Availability

The data presented in this study are available on request from the corresponding author. The data are not publicly available due to patients’ privacy and medical ethics.

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
