# Peer review of "Skin Surface Dose for Whole Breast Radiotherapy Using Personalized Breast Holder: Comparison with Various Radiotherapy Techniques and Clinical Experiences"

_cancers, 2022, doi:10.3390/cancers14133205_

Round 1

Reviewer 1 Report

The manuscript entitled "Skin Surface Dose for Whole Breast Radiotherapy Using Personalized Breast Holder: Comparison with Various Radiotherapy Techniques" details an interesting 3D printed breast holder and provides initial results from a feasibility study with 25 patients. Although, this is a novel research topic, the authors need to add more details and address some fundamental concerns before publication. Please see the detailed comments below:

1. Did the authors consider other materials for 3D printing instead of TPE ? Why was TPE chosen ? Did the authors perform a comparison of surface dose buildup - estimates or Monte Carlo study - to figure out whether TPE would be the optimal choice? Seems like it was only picked for it's rigidity. 

2. Lines 129-130 and 135-136 are repetitive

3. Line 145 - since these are female patients, please use "her" or "their" instead of his.

4. Line 149 - What dedicated image processing system do the authors use? Please provide more details.

5. Section 2: How were the gafchromic film and TLDs calibrated ? This needs to be added in the methods section. 

6. Line 235: Please provide more details of how the film was placed on the patients. What size were the cutouts, how were they attached, etc.
How would the smaller sized sections of film affect the accuracy of the dose readings ? 

7. Line 333: Please correct this grammatically incorrect sentence to "skin toxicity caused by wearing the PERSBRA."

8. Have the authors considered reducing the thickness of the PERSBRA from 5mm to something lower to reduce the surface dose buildup ? 

9. Details on time taken to 3D print the PERSBRA for each patient is an important factor missing from the manuscript. Readers need to understand how much more time will be added to standard treatment protocols by adding the different steps required here for each patient. 

10. The cost aspect should be considered. How much does each PERSBRA cost based on material used ? ANother important consideration for clinics that might want to consider incorporating this technique into their standard procedures. 

11. Finally, please add to the discussion what the authors think are shortcomings of the current study, hurdles that one would face in using this technique clinically and future research plans. 

Reviewer 2 Report

The article focus on breast radiotherapy using personalized breast holder. The technology used to treat patients who can not hold deep breath hold for more than 30 seconds. The results of manuscript provide advance treatment methods for breast cancer patients. 

I recommend this manuscript for publication in its current form. 

Reviewer 3 Report

The authors compared dosimetric results between PERSBRA with different size of mesh. In conclusion, the large mesh PERSBRA can be recommended for clinical use.

However, it looks the solid PERSBRA was applied to participating patients.

It could be an issue of ethical concern, I think.

Round 2

Reviewer 3 Report

I still do not understand why you include human data in this paper.

 The purpose of this study was to compare size of holes which may reduce bolus effect of PERSBRA. 

 Also, data about dose reduction for normal tissues were analyzed in the Rando phantom.

Author Response

Thank you for this opportunity to revise our Manuscript ID: 1736988 and Manuscript titled " Skin Surface Dose for Whole Breast Radiotherapy Using Personalized

Breast Holder: Comparison with Various Radiotherapy Techniques. Please see our point-by-point response to the reviewers below.

Reviewer Comments:

Reviewer #3:

I still do not understand why you include human data in this paper.

  The purpose of this study was to compare size of holes which may reduce bolus effect of PERSBRA. 

Response:

PERSBRA is a novel technique to reduce the dose in the heart with supine position, providing a similar effect to the prone position to keep away the distance from the heart to the treatment field. Through the rigorous IRB examination of Taipei Medical University, the consent of the patients was also obtained. Clinical data from patients is required to demonstrate the feasibility and clinical benefits of PERSBRA.

The reason we joined Human Body Data in this study is because we believe that Human Body Data can help readers understand the usefulness and potential of PERSBRA technology. However, we also agree with the reviewers that the inclusion of human data would obscure the focus of this study on surface dosimetry and we revise the title of this paper to make the reader more aware of the clinical benefits of PERSBRA and the comparison of surface doses.

Change in Manuscript:

Please see line 2-4 : Skin Surface Dose for Whole Breast Radiotherapy Using Personalized Breast Holder: Comparison with Various Radiotherapy Techniques and clinical experiences

Reference:

  1. TMU-Joint Institutional Review Board-N201603037.

Also, data about dose reduction for normal tissues were analyzed in the Rando phantom.

Response:

Please see line 254-260: Because the commonly used radiation therapy techniques for breast cancer are Hybrid, IMRT and VMAT. It is impossible to detect the real dose of radiation in the heart or lung of the patients. That is why we evaluates these three different techniques using PERSBRA to simulate normal tissues (lung, heart) doses. The readers can clearly understand the dose of normal tissues in different breast cancer radiation therapy techniques with PERSBRA.
